

# Exploring the systemic impacts of urinary tract infection-specific antibiotic treatments on the gut microbiome, metabolome, and intestinal morphology in rats

Hao Guo[1,*], Xiang Zhou[1,*], Zhou Li[1], Junjie Zhi[1], Chaowei Fu[2], Xinwei Liu[2], Yufan Wu[3], Fengping Liu[4] and Ninghan Feng[1]

[1] Department of Urology, Affiliated Wuxi No. 2 Hospital, Nanjing Medical University, Wuxi, China
[2] Jiangnan University Medical Center, Wuxi School of Medicine, Jiangnan University, Wuxi, China
[3] Department of Urology, Kunshan Hospital of Traditional Chinese Medicine, Kunshan, China
[4] Department of Wuxi School of Medicine, Jiangnan University, Wuxi, China
[*] These authors contributed equally to this work.

## ABSTRACT

The gut microbiota is a dynamic ecosystem that plays a crucial role in host metabolism, immune system regulation, and protection against pathogens. An imbalanced gut microbiota has been associated with various diseases such as inflammatory bowel disease, metabolic disorders, and autoimmune diseases. Antibiotic therapy can disrupt the balance of the microbiome, making it essential to understand the specific effects of these antibiotics on gut microbiota and related host function. This study aims to systematically investigate the impact of UTI-specific antibiotic treatment on the gut microbiota of rats. By analyzing changes in microbial composition and their effects on host metabolism and intestinal structure, we seek to provide new insights into the broader consequences of antibiotic intervention on host-microbiota interactions. The research findings indicate that antibiotic treatment leads to a significant disruption in microbial diversity and metabolic characteristics, as well as notable histological changes in the intestinal mucosa. These results highlight the need for cautious use of antibiotics, considering their extensive effects beyond antimicrobial activity. Future research should focus on strategies to mitigate these impacts, potentially through targeted antibiotic therapies or probiotics, to better balance treatment efficacy with health preservation.

## INTRODUCTION

A balanced microbiota plays a crucial role in maintaining overall health through its involvement in various physiological processes (*Zheng, Liwinski & Elinav, 2020*). The gut microbiota, in particular, functions as a dynamic ecosystem that contributes to host metabolism (*Lee & Hase, 2014*), immune system modulation (*Kamada et al., 2013a*; *Kamada et al., 2013b*), and protection against pathogens (*Kamada et al., 2013a*; *Kamada*

Corresponding authors
Fengping Liu, liu-laoshiyc@jiangnan.edu.cn
Ninghan Feng, n.feng@njmu.edu.cn

*et al., 2013b*). Disruptions in the balance of gut microbiota, known as dysbiosis, have been implicated in various diseases (*Hou et al., 2022*), including inflammatory bowel diseases (*Lane, Zisman & Suskind, 2017*), metabolic disorders (*Akash et al., 2019*), and autoimmune conditions (*Zhang et al., 2020*). Therefore, maintaining a diverse and balanced microbiota is essential for optimal health and well-being.

Antibiotic therapy, while essential for managing and eradicating bacterial infections, often targets both pathogenic and beneficial microbial residents of the gut indiscriminately. This broad-spectrum action can result in a state of dysbiosis (*Lankelma et al., 2017*; *Lankelma et al., 2016*; *Reese et al., 2018*), characterized by a significant reduction in microbial diversity and alterations in the community structure (*Dethlefsen & Relman, 2011*). Such changes can have cascading effects on the host, including increased susceptibility to infections, alterations in metabolic processes, and the development of chronic diseases such as obesity, diabetes, and inflammatory bowel disease (*Cho & Blaser, 2012*).

The consequences of antibiotic use extend beyond immediate health concerns. The long-term impacts on the microbiome can alter host-microbe interactions in ways that are not yet fully understood. For example, antibiotics can affect the metabolic capabilities of the microbiome, altering the production and absorption of key nutrients and metabolites (*Vliex et al., 2024*). These alterations can affect the immune response and the overall energy balance of the host (*Morgun et al., 2015*).

In clinical settings, antibiotics are particularly crucial for the treatment of urinary tract infections (UTIs), which are among the most common bacterial infections requiring medical intervention (*Behzadi et al., 2010*). The antibiotics used for UTI treatment, such as fluoroquinolones and beta-lactams, are known for their efficacy in eliminating urinary pathogens. However, their impact on the gut microbiome and the broader implications for the host's health are less well-documented. Given the frequency of UTI treatments, understanding the specific effects of these antibiotics on the gut microbiome and associated host functions is of paramount importance.

This study aims to systematically explore the ramifications of UTI-specific antibiotic treatments on the gut microbiome of Sprague-Dawley rats, a well-established model for human health research. By dissecting the changes in microbial composition and their subsequent effects on host metabolism and intestinal structure, this research seeks to provide new insights into the broader consequences of antibiotic interventions on host-microbial interactions.

## METHODS

### Animals and antibiotics treatment

In our study, we selected male Sprague-Dawley rats (8 weeks old, approximately 280 g) to minimize hormonal fluctuations that could influence the gut microbiota (*He et al., 2021*), thereby ensuring the consistency and reliability of our results. A one-week acclimatization period preceded the experimental protocol. The rats were obtained from the Animal Center of Jiangnan University, which follows strict ethical guidelines for animal care

(20201230S1200430[372]). Antibiotics for UTI treatment were administered to the experimental group (Abx) *via* oral gavage (*Liu et al., 2023*). The antibiotic cocktail was prepared by dissolving fosfomycin, nitrofurantoin, gentamicin, cefotaxime, rimantadine, and metronidazole each at a concentration of one g/L in the drinking water, resulting in a total antibiotic concentration of six g/L (*Naber et al., 2001*). Rats were divided into two groups: The Abx group received the antibiotic solution daily at a volume of two mL per rat by oral gavage to ensure accurate and consistent dosing. The control group (CN) received water. Both groups were dissected concurrently on the second weekend post-treatment initiation.

Animal care included daily monitoring for health and well-being, feeding with a standard laboratory diet, and housing in a temperature-controlled environment with a 12-hour light/dark cycle. Enrichment was provided through environmental stimuli such as nesting materials.

## Sample collection and procession

Prior to antibiotic administration, urine samples were obtained using metabolic cages. During the experimental interventions, no analgesia was given as the procedures were non-invasive and did not cause noticeable pain or distress. The following day, rats were euthanized with anesthesia. Post-induction of unconsciousness, they were disinfected with 1% iodine and subjected to a midline abdominal incision to collect 3–5 mL of blood from the inferior vena cava for hepatic and renal function assessments.

Euthanasia was performed using a two-step method: initial sedation with isoflurane followed by cervical dislocation, ensuring rapid and humane termination. Fecal samples were extracted from the transverse colon using sterile forceps and placed in sterile centrifuge tubes. For bacterial DNA analysis, samples were immediately treated with 1.5 mL of lysis buffer. For metabolite analysis, samples were flash-frozen in liquid nitrogen for 5 min, followed by storage at $-80\ °C$.

Criteria for euthanizing animals prior to the planned end of the experiment included signs of severe distress, significant weight loss (>20%), or unresponsive behavior. However, none of the animals met these criteria during the study.

At the conclusion of the experiment, surviving animals were either transferred to other ongoing research projects, adhering to ethical guidelines, or kept for further observation under standard laboratory conditions.

## Microbiome detection and analysis
### Bacterial DNA isolation and sequence

Fecal samples underwent 30 freeze-thaw cycles using a lysis buffer provided by Guhe Bio-Tech Co., Ltd. (Hangzhou, China). Samples were stored at $-80\ °C$ for no more than two weeks before DNA extraction. Genomic DNA was extracted using the (specific name) DNA Extraction Kit (Guhe Bio-Tech Co., Ltd., Wuhan, China), following the manufacturer's instructions. The purity and concentration of the extracted DNA were determined using a NanoDrop ND-1000 spectrophotometer (Thermo Fisher Scientific, Waltham, MA, USA), ensuring A260/A280 ratios between 1.8 and 2.0, and validated through 2% agarose gel electrophoresis.

The V4 region of the bacterial 16S rRNA gene was PCR-amplified using universal primers 515F (5′-GTGCCAGCMGCCGCGGTAA-3′) and 806R (5′-GGACTACHVGGGTWTCTAAT-3′). PCR amplification was performed under the following conditions: initial denaturation at 95 °C for 3 min; followed by 30 cycles of denaturation at 95 °C for 30 s, annealing at 55 °C for 30 s, and extension at 72 ° C for 30 s; with a final extension at 72 °C for 5 min. PCR products were purified using Agencourt AMPure XP Beads (Beckman Coulter, Brea, CA, USA) and quantified using the PicoGreen dsDNA Assay Kit (Thermo Fisher Scientific, Waltham, MA, USA).

Amplicons were pooled in equimolar concentrations to construct sequencing libraries. The library's size and concentration were assessed using an Agilent 2100 Bioanalyzer (Agilent Technologies, Santa Clara, CA, USA) and the Library Quantification Kit for Illumina (Kapa Biosystems, Wilmington, MA, USA), respectively. Sequencing was performed on the Illumina NovaSeq 6000 platform (Illumina, San Diego, CA, USA) at Guhe Bio-Tech Co., Ltd., generating paired-end 250 bp reads. Each sample yielded an average of approximately 100,000 high-quality sequences, ensuring adequate coverage for downstream analyses.

### Bioinformatic analysis

Raw 16S rRNA gene sequences were initially processed with Cutadapt (v2.10) to remove barcodes and adaptors. Overlapping paired-end reads were merged into longer contiguous sequences using FLASH (v1.2.8). Quality trimming of the reads was performed using fdtrim (v0.94), discarding bases with Phred quality scores below 20 from the 3′ end. Reads shorter than 100 bp, containing more than 5% ambiguous bases (Ns), or with an average quality score below 20 were excluded from further analysis. Chimeric sequences were identified and removed using VSEARCH (v2.3.4).

Clean reads were processed using QIIME 2 (v2020.6) to generate an amplicon sequence variant (ASV) table. ASVs were identified using the DADA2 algorithm within QIIME 2, which models and corrects Illumina-sequenced amplicon errors, providing higher resolution than traditional operational taxonomic unit (OTU) clustering methods. Taxonomic classification of ASVs was performed using a naïve Bayes classifier trained on the SILVA 16S rRNA gene reference database (release 138) specific to the V4 region. To assess the adequacy of sequencing depth, Good's coverage was calculated for each sample, with values exceeding 99%, indicating sufficient sampling of the microbial communities. Alpha diversity metrics, including Shannon and Simpson indices, were computed to evaluate within-sample diversity, while beta diversity was assessed using Bray-Curtis dissimilarity and visualized through principal coordinates analysis (PCoA). Statistical analyses were performed using R software (v4.0.3), employing the vegan and phyloseq packages for ecological and community analyses.

## Metabolites detection and analysis
### Metabolite extraction for LC-MS/MS analysis

Metabolites were extracted from 100 mg of fecal samples for untargeted metabolomics analysis using high-performance liquid chromatography coupled with quadrupole time-of-flight mass spectrometry. Samples were ground in liquid nitrogen, and the resulting

homogenate was resuspended in pre-chilled 80% methanol, followed by vigorous vortexing. The mixtures were then incubated on ice for 5 min and centrifuged at 15,000 g at 4 °C for 20 min. A portion of the supernatant was diluted to a final concentration containing 53% methanol using LC-MS grade water. This solution was transferred to a fresh Eppendorf tube and centrifuged again under the same conditions. The clarified supernatant was then injected into the LC-MS/MS system for analysis.

UHPLC-MS/MS analyses were conducted using a Vanquish UHPLC system coupled with either an Orbitrap Q Exactive™ HF or an Orbitrap Q Exactive™ HF-X mass spectrometer (Thermo Fisher Scientific, Waltham, MA, USA). The analyses were performed at Novogene Co., Ltd. (Beijing, China). Samples were separated on a Hypersil Gold column (100 × 2.1 mm, 1.9 μm) using a 12-minute linear gradient at a flow rate of 0.2 mL/min. The mobile phases consisted of eluent A (0.1% formic acid in water) and eluent B (methanol). The gradient profile was as follows: start at 2% B for 1.5 min; ramp to 85% B over 3 min; increase to 100% B over 10 min; return to 2% B over 10.1 min; hold at 2% B for an additional 12 min.

The Orbitrap Q Exactive™ HF mass spectrometer was operated in both positive and negative ionization modes. The operating parameters included a spray voltage of 3.5 kV, capillary temperature of 320 °C, sheath gas flow rate of 35 psi, auxiliary gas flow rate of 10 L/min, S-lens RF level of 60, and auxiliary gas heater temperature of 350 °C.

### Data processing and metabolite identification

The raw data files from the UHPLC-MS/MS analyses were processed using Compound Discoverer 3.3 (CD3.3, Thermo Fisher Scientific, Waltham, MA, USA) for peak alignment, peak picking, and quantitation of each metabolite. Key settings included a mass tolerance of 5 ppm, signal intensity tolerance of 30%, and a minimum intensity threshold. Initial peak areas were corrected using the first quality control (QC) sample. Subsequently, peak intensities were normalized to the total spectral intensity. This normalized data facilitated the prediction of molecular formulas, incorporating additive ions, molecular ion peaks, and fragment ions. Peaks were then compared against the mzCloud, mzVault, and MassList databases for accurate qualitative and quantitative analysis.

Statistical analysis was conducted using R (version 3.4.3), Python (version 3.12), and CentOS (release 6.6). For data not normally distributed, normalization was performed using the formula: raw quantitation value of the sample/(sum of sample metabolite quantitation values/sum of QC1 sample metabolite quantitation values), to calculate relative peak areas. Compounds with coefficient of variation (CV) in QC samples exceeding 30% were excluded to ensure robust metabolite identification and relative quantification.

Metabolites were annotated using the KEGG database, HMDB database, and LIPIDMaps database. Data analysis including principal components analysis (PCA) and partial least squares discriminant analysis (PLS-DA) was performed using metaX, a comprehensive software tailored for metabolomics data processing, and Permutation testing was employed to ensure the model's robustness and predictive power. Univariate analysis ($t$-test) was applied to assess statistical significance, with a threshold $P$-value of $< 0.05$. Metabolites with a variable importance in projection (VIP) score greater than 1, a $P$-value of $< 0.05$,

and a fold change (FC) of either $\geq 2$ or $\leq 0.5$ were classified as differential metabolites. Volcano plots, generated using ggplot2 in R, visualized these metabolites based on log2(Fold Change) and $-$log10($P$-value).

Heatmaps of differential metabolites were created after normalizing data using z-scores of intensity areas, utilizing the Pheatmap package in R. Correlations among differential metabolites were analyzed using the cor() function (method = pearson) in R, with significance determined by cor.mtest(), considering a $P$-value < 0.05 as statistically significant. Correlation plots were generated using the corrplot package in R.

The functions and pathways involving these metabolites were explored using the KEGG database. Enrichment of metabolic pathways among differential metabolites was assessed where the ratio satisfied x/n > y/N, and pathways were considered statistically significantly enriched at a $P$-value < 0.05.

## Growth, development, intestinal morphology

We employed metabolic cages to monitor water consumption, food intake, urine output, and fecal production of rats in both the Abx and control groups.

The jejunum samples were preserved overnight in 4% paraformaldehyde, rinsed in PBS, and subsequently subjected to a graduated ethanol dehydration process. Once the ethanol was replaced with xylene, the tissues were embedded in paraffin. The paraffin-embedded tissues were then cut into 5-$\mu$m sections, rehydrated in a descending ethanol series, immersed in distilled water, and stained with hematoxylin and eosin. Histological assessments were conducted by two pathologists who were unaware of the experimental setup. The structural integrity of the intestinal wall was evaluated by a blinded reviewer according to a modified version of *Chiu et al.'s (1970)* grading system: Grade 0 indicates normal mucosa; Grade 1 indicates the presence of a sub-epithelial gap at the villi tips; Grade 2 shows a wider sub-epithelial space and the formation of Gruenhagen's space at the villi tips; Grade 3 involves extensive epithelial lifting along the villi, with villus necrosis; Grade 4 is characterized by the complete loss of the epithelial layer on the villi; Grade 5 denotes total villous loss, mucosal ulceration, necrosis, and invasion into the muscularis propria. We utilized metabolic cages to gather information on water consumption, food intake, urine volume, and feces output of rats in the groups of Abx and controls.

## Urinary antimicrobial peptide measurement

Urinary antimicrobial peptide (AMP) concentrations were quantified using a competitive inhibition enzyme-linked immunosorbent assay (ELISA), conducted by Wuhan YourSun Biological Testing Center Co., Ltd. (Wuhan, China). The assay was based on a competitive binding mechanism, where sample-derived peptides competed with enzyme-labeled peptides for binding to immobilized antibodies.

## Statistical analysis

We employed Pearson's chi-square or Fisher's exact tests for categorical variables. Student's $t$-test was applied to normalized continuous variables, while the Wilcoxon rank-sum test was utilized for non-normally distributed continuous variables. To account for multiple comparisons, the $P$-values were adjusted using the Benjamini–Hochberg (BH) false

discovery rate (FDR). The significance threshold was established at an FDR-corrected value of < 0.05. The correlation between bacterial genera and metabolite was conducted by Spearman's correlation using the R package rstatix V.0.6.0 and correlation V.0.6.1.

## RESULTS

### Antibiotic-induced alterations in the gut microbiome of rats

In total, 16,258,893 high-quality sequences were retrieved from 18 rat gut samples, with an average of approximately 903,272 sequences per sample, achieving 99.84% Goods coverage, indicating that the sequencing depth was sufficient to capture the biodiversity of the samples comprehensively.

Principal coordinates analysis (PCoA) using the Bray Curtis distance algorithm highlighted a distinct separation in the bacterial community between the antibiotic-treated rats (Abx) and the control group, with significant differences noted ($P_{adj} < 0.05$; Fig. 1A). The analysis identified 296 ASVs, with 42 of these (14.19%) shared between the Abx and control groups (Fig. 1B). The Abx group showed a significant reduction in bacterial richness and diversity, as indicated by lower Chao 1, Shannon, and Simpson indices ($P_{adj} < 0.05$; Figs. 1C–1E).

In terms of bacterial phyla, the Abx rats had higher proportions of Proteobacteria and lower proportions of Firmicutes and Actinobacteriota compared to the control group, which was dominated by Firmicutes, Verrucomicrobiota, and Bacteroidota (Fig. 1F & Fig. S1). Notably, there were significant decreases in Bacteroidota, Desulfobacterota, Patescibacteria, and Verrucomicrobiota in the Abx group, whereas Proteobacteria levels decreased significantly in the control group ($P_{adj} < 0.05$; Fig. 1G).

Examining bacterial genera, the Abx group was primarily composed of *Escherichia-Shigella*, *Enterococcus*, and *Rothia*, while the control group featured *Akkermansia* and *Lactobacillus* as the predominant genera (Fig. 1H & Fig. S1). A comparative analysis of bacterial genera representing more than 0.5% of the total abundance revealed significant reductions in the Abx group for genera such as *Akkermansia*, Christensenellaceae R7 group, Clostridia_UCG-014, Eubacterium coprostanoligenes group, NK4A214 from Oscillospiraceae, UCG-005 from Oscillospiraceae, Lachnospiraceae NK4A136 group, *Muribaculum*, and *Ruminococcus*. Conversely, there were notable increases in *Enterococcus* and *Escherichia-Shigella* among the Abx rats ($P_{adj} < 0.05$; Fig. 1I).

### Antibiotic-induced alterations in the gut metabolites of rats

To assess the metabolic changes induced by antibiotic administration, we employed untargeted LC-MS metabolomics to analyze the metabolomic profiles of fecal samples from both Abx and control rats. Orthogonal projections to latent structures discriminant analysis (OPLS-DA) and PCA distinctly separated the Abx rats from the controls, highlighting significant metabolic differences (FC > 3, VIP > 1; Figs. 2A–2B). A volcano plot demonstrated that 160 metabolites were down-regulated, and 227 were up-regulated in the Abx rats compared to the controls ($P_{adj} < 0.05$; Fig. 2C). Among these, 11 metabolites, including Creatine and α-Aspartylphenylalanine, were significantly up-regulated, while

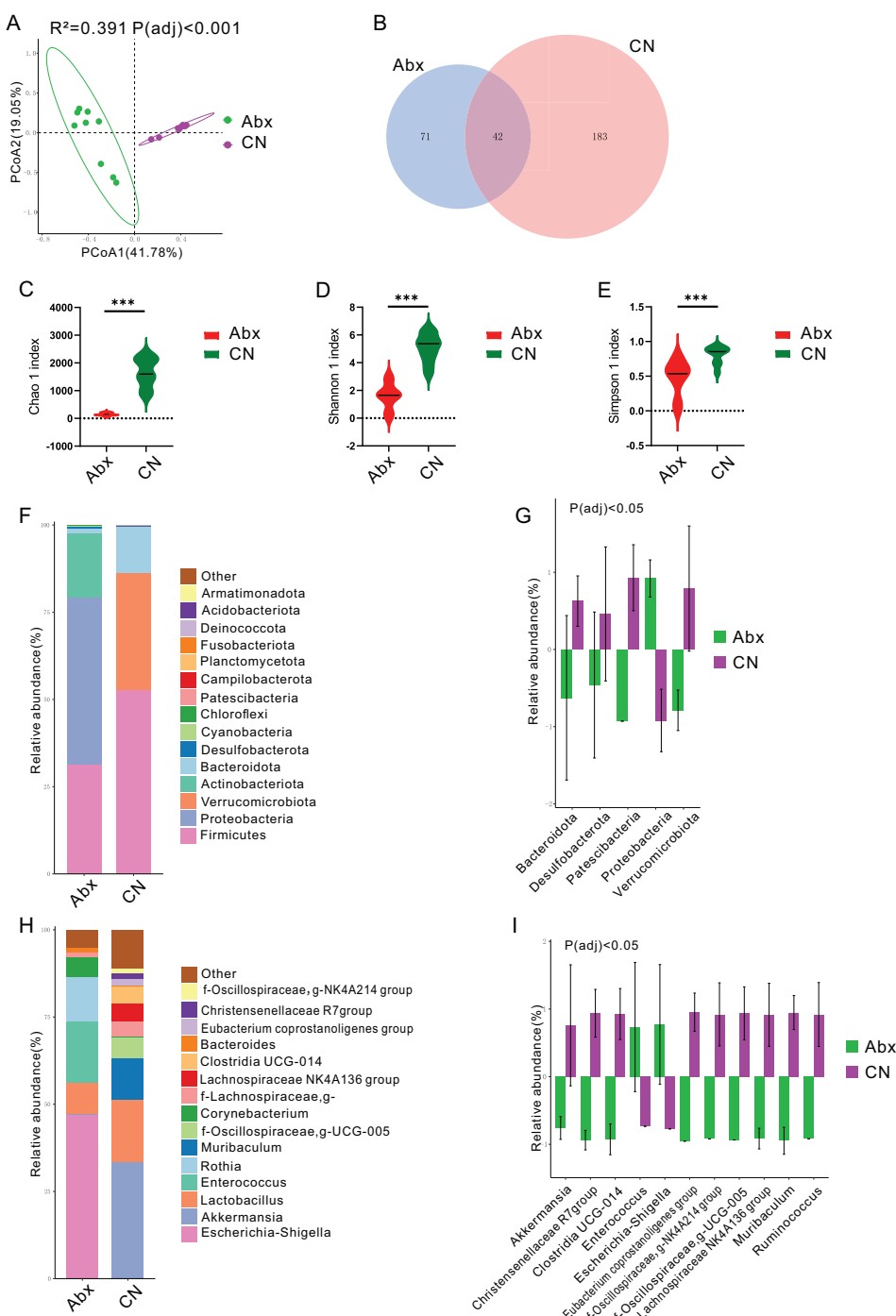

**Figure 1** **Impact of antibiotic treatment on gut microbiota composition.** (A) Principal Coordinates Analysis (PCoA) plot showing significant differentiation in gut microbiota composition between Abx and control groups, with 41. 78% variation explained by PCoA1 and an adjusted $P_{adj} < 0.001$; (B) Venn diagram depicting the unique and shared ASVs between the Abx and control groups; (C–E) Violin plots illustrating significantly higher Chao1, Shannon and Simpson index (continued on next page…)

**Figure 1 (...continued)**
values in the control group compared to the Abx group, indicating greater microbial richness (***$P_{adj} <$ 0.001); (F) Bar chart of relative bacterial phylum-level abundance in the Abx and CN groups, showing major shifts in microbial populations due to antibiotic treatment; (G) Bar chart comparing the relative abundance of key bacterial families between the Abx and control groups, with statistical differences noted where $P_{adj} < 0.05$; (H) Bar chart of relative abundance at the genus level within the gut microbiota of the Abx and control groups; (I) Bar chart detailing the relative abundance of selected bacterial genera showing significant differences between the Abx and control groups, with statistical differences noted where $P_{adj} < 0.05$.

33 metabolites, such as 1-Methylguanine, 3-Hydroxyproline, and D-Ribose, were down-regulated when meeting criteria of FC > 3 and VIP > 1 ($P_{adj} < 0.05$; Fig. 2D). Subsequent KEGG enrichment analysis identified affected metabolic pathways in the gut samples, including bile secretion, ABC transporters, protein digestion and absorption, glyoxylate and dicarboxylate metabolism, and the metabolism of glycine, serine, and threonine, providing insights into the biochemical impact of antibiotics on the gut metabolome of rats (Fig. 2E).

## Correlation between altered bacterial genera and betabolites

We performed Pearson correlation analysis to explore the relationships between differentially abundant bacterial genera and altered fecal metabolites in the antibiotic-treated and control rats. The analysis revealed several significant correlations: *Escherichia Shigella* showed a positive correlation with various metabolites including creatine, N1-(1-benzyl-4-piperidyl)-4-chlorobenzene-1-sulfonamide, FQH, and oxytetracycline. Conversely, *Akkermansia* and *Muribaculum* displayed negative correlations with D-Ribose. Enterococcus was positively correlated with Tetrahydrocorticosterone ($P < 0.05$; Fig. 3).

## Urinary antimicrobial peptides remain unchanged following antibiotic treatment

To explore whether gut microbiota dysbiosis affects host defense in the urinary tract, we measured the concentration of antimicrobial peptides (AMPs) in urine samples from both the Abx and control groups. Although a decreasing trend was observed in the Abx group, there was no statistically significant difference in urinary AMP levels between the two groups ($P = 0.841$; Table S1). These results suggest that short-term antibiotic-induced alterations in gut microbiota may not immediately impact AMP secretion in the urinary tract.

## Impact of UTI antibiotics on intestinal mucosal histomorphology

To explore the potential effects of UTI-specific antibiotic treatment on intestinal mucosal histomorphology, we conducted histological analyses on the jejunum segments extracted from treated rats. Compared to the control group, the antibiotic-treated rats displayed marked alterations in the intestinal architecture. Notably, there was significant evidence of necrosis in the intestinal villi and other structural deformities, which were statistically significant ($P < 0.05$; Figs. 4A–4C).

Peer

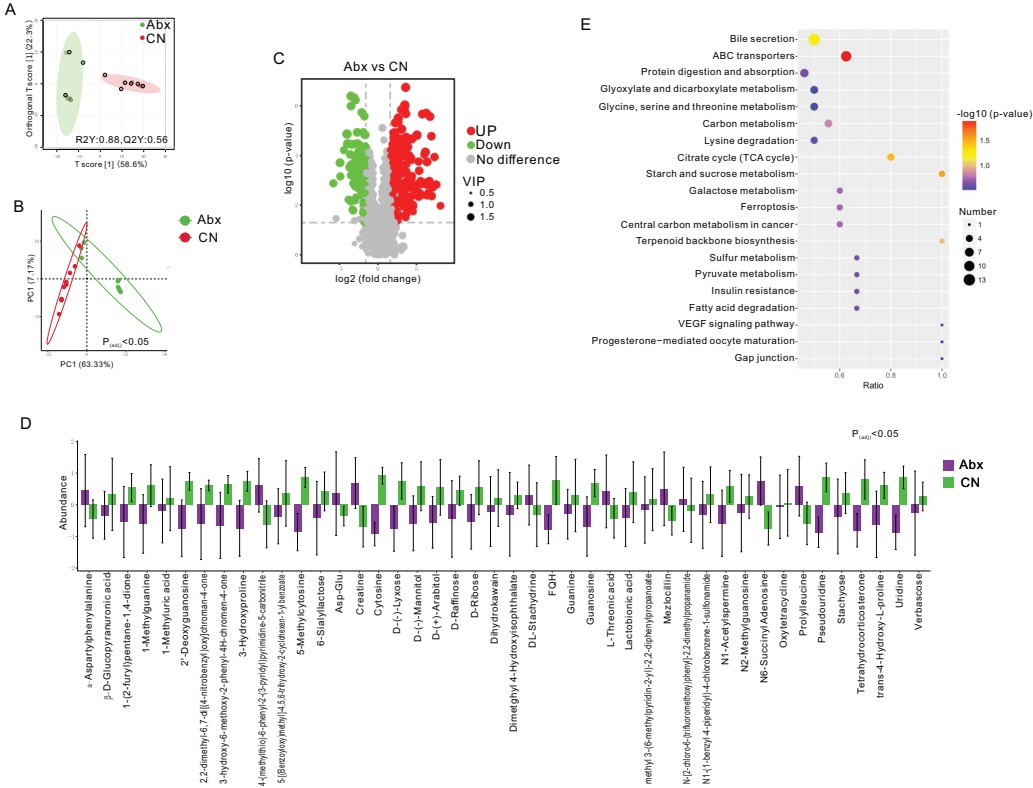

**Figure 2** **Comparative analysis of fecal metabolites between Abx and control groups.** (A) Orthogonal partial least squares discriminant analysis (OPLS-DA) score plot showing the separation between the Abx and CN groups based on fecal metabolites. The ellipses represent the 95% confidence intervals; (B) Principal component analysis (PCA) indicating significant differences between the groups ($P_{adj} < 0.05$), with PC1 accounting for 63.33% of the variation; (C) Volcano plot highlighting the significantly upregulated (green) and downregulated (red) metabolites in the Abx group compared to the CN group. Gray dots represent metabolites with no significant change; (D) Bar graph of the relative abundance of various metabolites in the Abx and control groups, demonstrating significant variability in metabolite levels; (E) Pathway enrichment analysis showing the metabolic pathways impacted by antibiotic treatment, with the size and color of the dots indicating the number of metabolites involved and the significance of pathway enrichment, respectively.

## Impact of UTI antibiotics on rat growth parameters

In our study, we monitored growth parameters including body weight, as well as food and water intake, alongside feces and urine output among rats in the Abx and the control group. Our analysis revealed a statistically significant decrease in food and water consumption, and reduced fecal and urine output in the rats receiving antibiotics compared to those in the control group ($P < 0.05$; Fig. 5).

## DISCUSSION

This study investigated the effects of antibiotic treatment on the gut microbiome, metabolome, and intestinal morphology of Sprague-Dawley rats, revealing significant alterations that highlight the profound impact of antibiotics beyond their intended

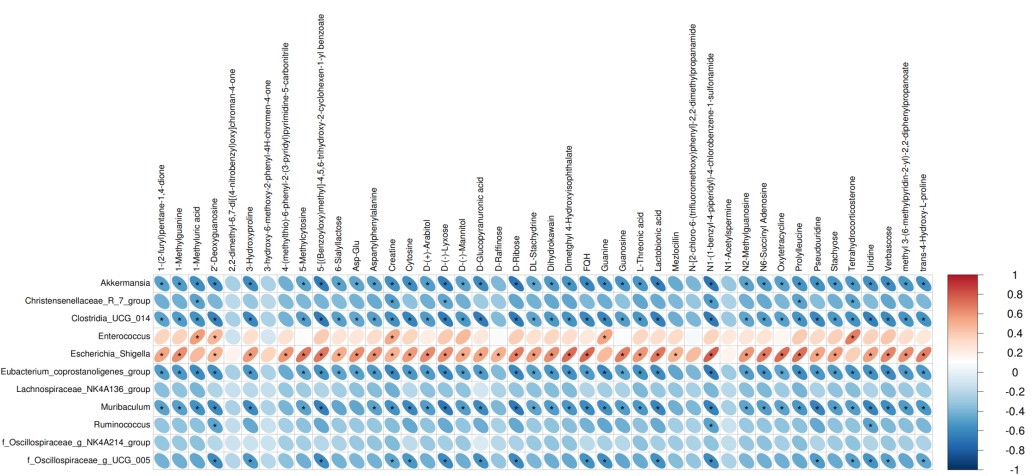

**Figure 3** **Correlations between microbiome and metabolites.** The heatmap presents the correlations between the differential bacterial genera and the differential metabolites between the groups of Abx and control. The legend on the right indicates the correlation coefficients: the redder the color, the stronger the positive correlation; the bluer the color, the stronger the negative correlation. The more elongated the ellipse, the higher the absolute value of the correlation. Areas marked with an asterisk (*) indicate $P \leq 0.05$. The Pearson statistical method is used to calculate the correlation coefficients ($r$) and $P$-values.

therapeutic targets. The findings demonstrated a marked disruption in the microbial diversity and metabolic profiles, coupled with notable histological changes in the intestinal mucosa. These changes align with the hypothesis that antibiotic treatment leads to a systemic alteration in the host's microbiological and metabolic environment, corroborating and expanding upon previous research that has outlined the broad-reaching effects of antibiotics on host health (*Dethlefsen & Relman, 2011*; *Cho & Blaser, 2012*).

The profound changes in the gut microbiome composition observed in this study, as evidenced by the distinct clustering in PCoA, confirm the significant impact of antibiotic treatment on microbial community structure. The significant reduction in microbial richness and diversity, as indicated by the Chao 1, Shannon, and Simpson indices, is consistent with previous findings that antibiotics reduce microbial diversity, which is crucial for maintaining gut homeostasis (*Xu et al., 2020*; *Doan et al., 2017*). Such reductions can lead to a weakened microbial ecosystem less capable of resisting colonization by pathogenic organisms, potentially increasing the host's susceptibility to infections and disease.

The observed increase in Proteobacteria and decrease in Firmicutes and Actinobacteriota in the antibiotic-treated rats are particularly concerning, given that an overrepresentation of Proteobacteria is often associated with dysbiosis and has been linked to various diseases, including inflammatory bowel disease (IBD) and metabolic syndrome. In contrast, Firmicutes and Actinobacteriota are typically involved in beneficial metabolic processes, such as fermenting dietary fibers to produce short-chain fatty acids that support gut health.

The dramatic shifts in bacterial genera, with increases in potentially pathogenic genera like *Escherichia-Shigella* and *Enterococcus*, further underscore the disruptive impact of antibiotics. These genera are often associated with hospital-acquired infections and may

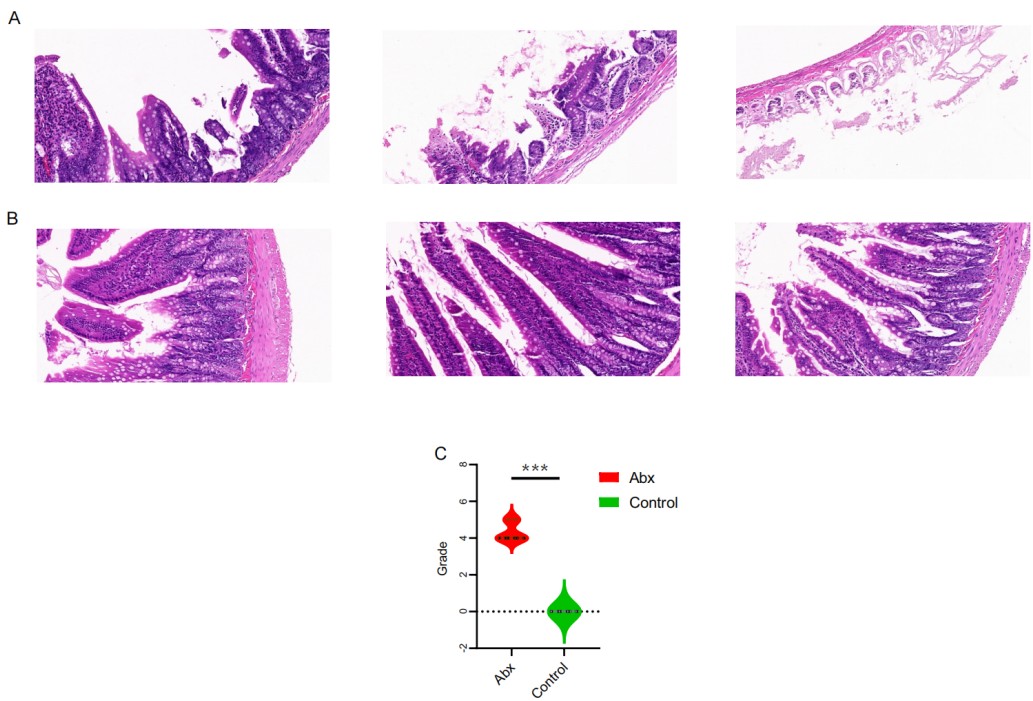

**Figure 4   Comparison of jejunal tissue morphology between antibiotic-treated and control rats.**
(A–B) Representative hematoxylin and eosin-stained sections of jejunum tissue from rats treated with
Abx and control rats; (C) comparison of histomorphological changes in the jejunum between Abx and
control groups. The data are presented as means ± SEM. Statistical significance is indicated with "***" for
$P < 0.001$, highlighting notable differences between the groups.

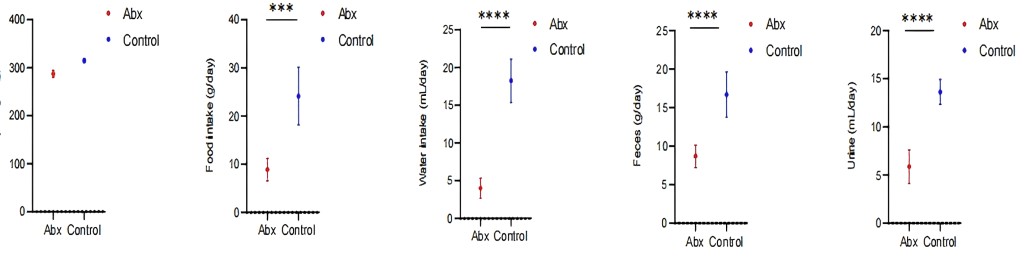

**Figure 5   Impact of UTI-specific antibiotics on growth parameters in rats.** Comparison of body weight,
food and water intake, and fecal and urine output between Abx and control groups. A $t$-test was employed
to evaluate statistical significance between the groups. The symbols "***" and "****" denote $P$-values of
less than 0.001 and 0.0001, respectively.

have pathogenic potential, underlining the risk of opportunistic infections following
antibiotic-induced dysbiosis (*Langdon, Crook & Dantas, 2016*). Conversely, the significant
reductions in beneficial genera such as *Akkermansia* and *Lactobacillus*—known for their
roles in maintaining mucosal integrity and inhibiting pathogen adhesion, respectively—are
notable (*Derrien et al., 2008*). The loss of these beneficial microbes could compromise gut
barrier functions and immune modulation.

The metabolomic analysis conducted in this study provides critical insights into the biochemical consequences of antibiotic administration on Sprague-Dawley rats. The application of untargeted LC-MS metabolomics successfully delineated the metabolic distinctions between antibiotic-treated rats and controls, as evidenced by the clear separation in OPLS-DA and PCA. These findings robustly highlighted the extensive metabolic reprogramming that antibiotics induce in the gut environment. The differential expression of 387 metabolites, with 227 up-regulated and 160 down-regulated, underscores the significant disruption caused by antibiotics. The substantial alteration in levels of key metabolites such as creatine and α-aspartylphenylalanine, which were significantly up-regulated, points to a shift towards altered energy metabolism and amino acid processing. Creatine, typically associated with energy storage and supply, may indicate a compensatory response to energy deficiency or a shift in energy metabolism pathways due to microbiome alterations (*Kazak & Cohen, 2020*). Similarly, the increase in α-aspartylphenylalanine suggests changes in protein metabolism (*Trawiński & Skibiński, 2023*), possibly reflecting an increased demand for specific amino acids during stress or inflammatory conditions induced by microbial imbalances. Conversely, the down-regulation of metabolites such as 1-methylguanine, 3-hydroxyproline, and D-ribose, which are crucial in nucleotide synthesis, collagen formation, and carbohydrate metabolism respectively (*Falnes, 2004*; *Weis et al., 2010*; *Chen et al., 2017*), suggests a disruption in these fundamental biological processes. This could potentially affect tissue repair processes, DNA synthesis, and energy production pathways, illustrating the far-reaching impacts of antibiotic treatments beyond their antimicrobial action.

The results of correlation analysis significantly illuminate the consequences of antibiotic treatment on the gut microbiota and subsequent metabolic dysregulation in Sprague-Dawley rats. Antibiotic treatment resulted in significant shifts in the abundance of key bacterial genera, which correlated strongly with changes in specific metabolites. *Escherichia Shigella*, typically resilient to antibiotic pressure, showed a positive correlation with metabolites such as creatine, N1-(1-benzyl-4-piperidyl)-4-chlorobenzene-1-sulfonamide (FQH), and oxytetracycline. This suggests that the altered gut environment favors the proliferation of potentially pathogenic bacteria like *Escherichia Shigella*, which may exploit the ecological niches vacated by antibiotic-sensitive bacteria. The association of these bacteria with diverse metabolites underlines the broad impact of microbial alterations on the host's metabolic pathways, particularly those related to energy metabolism and xenobiotic processing (*Oudman, Clark & Brewster, 2013*; *Qian et al., 2021*). Conversely, the negative correlations observed between *Akkermansia* and *Muribaculum* with D-ribose highlight a significant metabolic impact resulting from the loss of these beneficial microbes (*Lv et al., 2023*; *Huang et al., 2023*). Both genera are known for their roles in maintaining gut barrier integrity and metabolic homeostasis (*Lv et al., 2023*; *Huang et al., 2023*). Their reduction correlates with disturbances in carbohydrate metabolism (*Macchione et al., 2019*), evidenced by alterations in D-ribose levels, a crucial component in cellular energy processes (*Shecterle, Terry & St, 2018*). This aligns with findings that link reduced populations of these bacteria to compromised metabolic health and increased metabolic disease risk (*Dao et al., 2016*; *Zheng et al., 2022*). Furthermore, the positive correlation

between *Enterococcus* and tetrahydrocorticosterone reinforces the notion that microbial shifts influence not only metabolic but also physiological stress responses (*Scheun, Greeff & Ganswindt, 2018*). Elevated tetrahydrocorticosterone, associated with *Enterococcus* proliferation, could reflect a stress response to microbial imbalance, indicating a complex interplay between gut microbiota alterations and host adrenal functions. These correlations observed suggest that restoring a balanced microbiota could potentially reverse some of the metabolic dysfunctions induced by antibiotic treatments.

The histological findings from our study underscore the profound effects of UTI-specific antibiotic treatments on the intestinal mucosal histomorphology of Sprague-Dawley rats. These results, featuring significant necrosis and structural deformities in the intestinal villi, illustrate the severity of tissue damage that can be induced by such antibiotic interventions. Our findings link these structural changes to the above-mentioned shifts in microbial diversity and metabolic reprogramming. The reduction in beneficial bacteria such as *Lactobacillus* and the increase in pathogenic bacteria like *Escherichia-Shigella* might not only influence the metabolic processes but also exacerbate the vulnerability of the intestinal mucosa to damage (*Shi et al., 2021*; *Sun et al., 2019*). This suggests a bidirectional relationship where microbial dysbiosis contributes to mucosal damage, and conversely, mucosal damage could further disrupt the gut microbiome.

Our findings indicate that the rats receiving antibiotics showed a reduction in food and water intake, as well as decreased fecal and urine output. These changes in consumption and excretion patterns are likely intertwined with the disruptions previously noted in the gut microbiota and metabolic profiles (*Bongers et al., 2022*). The reduction in food and water intake could be a direct consequence of the dysbiosis and intestinal damage induced by the antibiotics. Damage to the intestinal villi, as detailed earlier, can impair nutrient absorption and digestion, leading to decreased appetite and altered metabolic needs (*Judkins et al., 2020*). Similarly, changes in the microbiota might impact the production of neurotransmitters and hormones involved in hunger signaling, further influencing feeding behavior (*Sun, Li & Nie, 2020*; *Smitka et al., 2021*). The decreased fecal and urine output observed may reflect disruptions in the gut's motility and a reduced efficiency in waste processing and excretion. Antibiotics can alter the gut's muscular contractions and the reabsorption of nutrients and water, leading to changes in stool consistency and urine production. These findings are consistent with the notion that antibiotics not only disrupt microbial ecosystems but also affect the physiological processes they regulate (*Willing, Russell & Finlay, 2011*).

There are several limitations should be considered for our present study. First, the use of Sprague-Dawley rats and a cocktail of multiple antibiotics may limit the direct applicability of the findings to human clinical settings, where single antibiotic treatments are more common, and physiological responses may differ significantly from those observed in rodents. Second, the study relies on 16S rRNA gene sequencing, which restricts the analysis to bacterial components of the microbiota, excluding viruses, fungi, and protozoa that might also play critical roles in the microbiome. Additionally, untargeted metabolomics might miss specific metabolites crucial for understanding metabolic alterations fully. Third, the experimental design captures only immediate post-treatment effects without

assessing the long-term recovery of microbial and metabolic changes. Moreover, potential variability in environmental factors such as diet and housing conditions, which can influence microbiota and metabolic outcomes, was not controlled rigorously. Fourth, while this study demonstrates associations between antibiotic exposure, gut microbiome alterations, and systemic changes in intestinal and metabolic profiles, it does not include a UTI model. Therefore, our conclusions regarding the potential link between gut dysbiosis and UTI susceptibility remain correlative. Future work incorporating UTI challenge models, along with longitudinal sampling of microbiota and host defense markers, will be critical to establish a causal relationship. Nonetheless, our current findings provide a foundation for understanding how UTI-targeted antibiotics may impact host-microbiota interactions beyond the urinary tract.

## CONCLUSION

In conclusion, this study demonstrated that UTI-specific antibiotic treatments in Sprague-Dawley rats significantly disrupt the gut microbiome, metabolome, and intestinal morphology. Our findings reveal that antibiotics not only reduce microbial diversity but also induce substantial metabolic reprogramming and damage to intestinal structures. These results highlight the need for cautious use of antibiotics, considering their extensive effects beyond antimicrobial activity. Future research should focus on strategies to mitigate these impacts, potentially through targeted antibiotic therapies or probiotics, to better balance treatment efficacy with health preservation.

## ACKNOWLEDGEMENTS

We extend our gratitude to the Department of Pathology at Affiliated Wuxi No. 2 Hospital for their invaluable assistance in the preparation of tissue sections.

### Funding

This work was supported by the National Natural Science Foundation of China (82370777, 81874142 and 82073041); Zhejiang Provincial Natural Science Foundation of China (LXR22H160001, and LY22H160011); Gusu Medical Talent Foundation (GSWS2020021). The funders had no role in study design, data collection and analysis, decision to publish, or preparation of the manuscript.

### Grant Disclosures

The following grant information was disclosed by the authors:
The National Natural Science Foundation of China: 82370777, 81874142, 82073041.
Zhejiang Provincial Natural Science Foundation of China: LXR22H160001, LY22H160011.
Gusu Medical Talent Foundation:  GSWS2020021.

### Competing Interests

The authors declare there are no competing interests.

## Author Contributions

- Hao Guo conceived and designed the experiments, performed the experiments, analyzed the data, prepared figures and/or tables, authored or reviewed drafts of the article, and approved the final draft.
- Xiang Zhou conceived and designed the experiments, performed the experiments, analyzed the data, prepared figures and/or tables, authored or reviewed drafts of the article, and approved the final draft.
- Zhou Li conceived and designed the experiments, performed the experiments, prepared figures and/or tables, authored or reviewed drafts of the article, and approved the final draft.
- Junjie Zhi conceived and designed the experiments, performed the experiments, prepared figures and/or tables, authored or reviewed drafts of the article, and approved the final draft.
- Chaowei Fu conceived and designed the experiments, performed the experiments, prepared figures and/or tables, and approved the final draft.
- Xinwei Liu conceived and designed the experiments, performed the experiments, prepared figures and/or tables, and approved the final draft.
- Yufan Wu conceived and designed the experiments, analyzed the data, authored or reviewed drafts of the article, and approved the final draft.
- Fengping Liu conceived and designed the experiments, analyzed the data, prepared figures and/or tables, authored or reviewed drafts of the article, and approved the final draft.
- Ninghan Feng conceived and designed the experiments, prepared figures and/or tables, authored or reviewed drafts of the article, and approved the final draft.

## Animal Ethics

The following information was supplied relating to ethical approvals (i.e., approving body and any reference numbers):

Animal Ethics Committee of Jiangnan University provided full approval for this research (20201230S1200430[372]).

## DNA Deposition

The following information was supplied regarding the deposition of DNA sequences:

The sequences are available at NCBI: PRJNA898681.

## Microarray Data Deposition

The following information was supplied regarding the deposition of microarray data:

The sequences are available at NCBI: PRJNA898681.

## Data Availability

The sequences are available at NCBI: PRJNA898681.

## Supplemental Information

Supplemental information for this article can be found online at http://dx.doi.org/10.7717/peerj.19486#supplemental-information.

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
