# Peer review of "Exploring the systemic impacts of urinary tract infection-specific antibiotic treatments on the gut microbiome, metabolome, and intestinal morphology in rats"

_PeerJ, doi:10.7717/peerj.19486_

## Round 0.1 · original submission · Major Revisions

Please address the reviewer comments and submit a revision of the manuscript.

Reviewer 1 ·

Basic reporting

he ethical approval reference (line 101) "20201230S1200430[372]" appears to have an unclear or incorrect format. It should be clearly referenced according to institutional or journal guidelines.

Experimental design

he manuscript states that the Abx group received the antibiotic solution daily, but it does not specify the volume administered per rat or how intake was monitored to ensure consistent dosing. The methods for bacterial DNA extraction mention a "freeze-thaw technique", but it would be more informative to briefly describe the number of freeze-thaw cycles and specific reagents used. Fecal samples were stored at -80°C (line 129), but there is no mention of how long they were stored before analysis, which can affect microbial DNA integrity.

Validity of the findings

The statement “893 sequences were retrieved from 18 rat gut samples” (line 243) is vague. The total number of sequences seems too low for a 16S rRNA sequencing study, where typical sequence counts are in the thousands or millions per sample. This should be clarified.The text states that Proteobacteria levels increased significantly in the control group, which contradicts the earlier statement (line 252) that Abx rats had higher proportions of Proteobacteria. This needs to be corrected for consistency. While 16S rRNA sequencing is mentioned, important details like sequencing depth per sample, quality control measures, and bioinformatics tools used for ASV assignment are missing.

Additional comments

I think that overall, the article is okay and could be published after minor changes.

·

Basic reporting

No comment

Experimental design

Ln 104, please indicate the specific concentration of each antibiotic in the cocktail.
The authors should indicate how UTI was induced in the rats and what was used as control for this.

Validity of the findings

The finding produced in the study are scientifically valid. This is due to the methodology used and the accurate description will enhance reproducibility.

Reviewer 3 ·

Basic reporting

1. Please provide higher magnification images of the jejunum in Figure 4.
2. In Figure 6, please modify the figure legend to discriminate between A and B.

Experimental design

1. Please explain rationale for using only male mice. Urinary tract infection is more common in females, as well as antibiotic use.
2. Please consider measuring host defense molecules in the urine collected from these mice. Given the focus on urinary tract infection in the manuscript, it would be interesting to see if, for instance, antimicrobial peptides are altered in the urinary tract as a result of gut microbiome dysbiosis. It is known that antimicrobial peptides play a key role in maintaining gut microbiome diversity, favoring growth of beneficial microbes and preventing overgrowth of pathogenic microbes, thus the impact of antibiotic use on these antimicrobial peptides would correlate changes in gut microbiome to infection risk.

Validity of the findings

The data is clearly presented and conclusions are well stated. However, the outcomes of this research are only correlative. The impact of changes in gut microbiome diversity as a result of this specific regimen of antibiotics is unknown, as no infection studies were carried out. The authors do state this explicitly in the discussion as a future direction. However, I think this limitation diminishes the impact of the work to a degree.

---

## Round 0.2 · Minor Revisions

Please answer the reviewer's last few comments.

Reviewer 1 ·

Basic reporting

no comment

Experimental design

no comment

Validity of the findings

no comment

Additional comments

After the extensive revision, I think that the editorial process can continue. However, I still have some minor comments:
Line 109- Why did you insert a link in the text?
Line 193 – Why did you use an ancient version of Python? Version 2.7 is end-of-life and no longer supported.
Line 199-200 - The protocol mentions using PLS-DA but doesn't specify if validation techniques (like permutation testing or cross-validation) were employed to ensure the model's robustness and predictive power.

Reviewer 3 ·

Basic reporting

No comment

Experimental design

No comment

Validity of the findings

No Comment

Additional comments

Thank you for adequately addressing reviewer comments with your revisions.

---

## Round 0.3 · accepted · Accept

Thaks for addressing all comments!

Reviewer 1 ·

Basic reporting

No comment

Experimental design

No comment

Validity of the findings

No comment

Additional comments

Dear authors, after reading the revised version, from my point of view, the editorial process can continue